# The Urban-Rural Disparity in the Status and Risk Factors of Health Literacy: A Cross-Sectional Survey in Central China

**DOI:** 10.3390/ijerph17113848

**Published:** 2020-05-29

**Authors:** Wenna Wang, Yulin Zhang, Beilei Lin, Yongxia Mei, Zhiguang Ping, Zhenxiang Zhang

**Affiliations:** 1School of Nursing and Health, Zhengzhou University, Zhengzhou 450001, China; wwn2436645698@gs.zzu.edu.cn (W.W.); linbeilei.com.cn@163.com (B.L.); myx@zzu.edu.cn (Y.M.); 2Institute of Health Education and Prevention of Chronic Non-Communicable Diseases, Henna Provincial Centers for Disease Control and Prevention, Zhengzhou 450016, China; 3School of Public Health, Zhengzhou University, Zhengzhou 450001, China; pingzhg@zzu.edu.cn

**Keywords:** health literacy, urban, risk factor, rural, China

## Abstract

Health literacy is the ability of individuals to access, process, and understand health information to make decisions regarding treatment and their health on the whole; it is critical to maintain and improve public health. However, the health literacy of urban and rural populations in China has been little known. Thus, this study aims to assess the status of health literacy and explore the differences of its possible determinants (e.g., socio-economic factors) among urban and rural populations in Henan, China. A cross-sectional study, 78,646 participants were recruited from a populous province in central China with a multi-stage random sampling design. The Chinese Resident Health Literacy Scale was adopted to measure the health literacy of the respondents. In the participants, the level of health literacy (10.21%) in central China was significantly lower than the national average, and a big gap was identified between urban and rural populations (16.92% vs. 8.09%). A noticeable difference was reported in different aspects and health issues of health literacy between urban and rural populations. The health literacy level was lower in those with lower levels of education, and a significant difference was identified in the level of health literacy among people of different ages and occupations in both urban and rural areas. Note that in rural areas, as long as residents educated, they all had higher odds to exhibit basic health literacy than those uneducated; in rural areas, compared with those aged 15 to 24 years, residents aged 45 to 54 years (OR = 0.846,95% CI (0.730, 0.981)), 55 to 64 years (OR = 0.716,95% CI (0.614, 0.836)) and above 65 years (OR = 0.679, 95% CI (0.567, 0.812)) were 84.6%, 71.6%, and 67.9%, respectively, less likely to exhibit basic health literacy. Considering the lower health literacy among rural residents compared with their urban counterparts, a reorientation of the health policy-making for Chinese rural areas is recommended. This study suggests that urban–rural disparity about health literacy risk factors should be considered when implementing health literacy promotion intervention.

## 1. Introduction

Health literacy is the ability of individuals to access, process, and understand health information to make decisions in terms of treatment and their health on the whole [1,2]. It is a strong predictor for health status and significantly impacts public health [3]. Existing studies suggested that high health literacy is associated with healthier lifestyles and health-promoting and -maintaining behaviors in healthy adults and people with chronic disease [4,5]. Meantime, increasing studies suggested that inadequate health literacy is associated with many negative health outcomes (e.g., poor self-rated health, occurrence and development of chronic diseases, and increased mortality risk) [6,7]. Health literacy also displays an inverse relationship to healthcare utilization and expenditure and critically impacts health education and promotion [8]. Accordingly, health literacy is increasingly important in public health.

Since health literacy affects most of the population [9], it has become more popularized over the past decade. Though China is a large country with a population of approximately 1.4 billion [10], compared with the long and comprehensive history of health literacy studies in different settings of some developed countries, (e.g., the United States or Canada), it has a late start in focusing on health literacy, and the term “health literacy” was proposed initially in the project of “the First Chinese Residents Health Literacy Monitoring Program” launched by China Center for Disease Control and prevention in 2005 and the first national survey of the public health literacy status was conducted by the Chinese Ministry of Health in the mainland of China in 2008 [11,12]. Subsequently, in 2012, the Chinese Health Education Authority issued a novel strategic plan to assess health literacy nationally; the result indicated that the health literacy monitoring work has ushered into the track of normalization and standardization [13]. As suggested by the national health literacy monitoring data, through continuous intervention, the national health literacy level of Chinese residents was steadily elevated from 6.48% in 2008 to 9.79% in 2014 [14]. However, the rapid and disproportional economic growth in urban and rural China has brought unexpected changes in public health [15]. One study conducted in the Jiangsu province of China in 2010 demonstrated that living in a rural area acts as a critical predictor of a low health literacy status [16], and a recent study conducted in Hebei Province, China has reported that that health literacy level of urban residents (19.00%) is significantly higher than that in rural areas (7.94%) [17]. Moreover, China has become an “aging society” as of 2000 and has experienced an unprecedented process in its aging population [18]. The accelerated population aging has brought about health, social, and economic issues (e.g., decreased health function attributed to disability and chronic diseases) [19], and note that the burden of chronic diseases is serious in rural China. Hence, elevating the health literacy level of residents and narrowing the gap between urban and rural health literacy is crucial to the steady improvement of public health in China.

An individual with an adequate level of health literacy refers to a person that has the ability to take responsibility for his or her own health, as well as their family’s health and community health [1]. Though adequate health literacy and knowledge about associated factors are relevant in all phases of the life course to maintain health and getting involved in decisions in terms of health and health care [9], health literacy status and its risk factors between urban and rural areas have been little known. To gain insights into the health literacy and deepen the implementation of health literacy intervention among residents from a regional perspective, the aim of this study is therefore to assess the status of health literacy and to explore the differences of its possible determinants (e.g., socio-economic factors) among urban and rural populations in Henan province, China.

## 2. Materials and Methods

### 2.1. Study Population and Sampling Design

This study was a cross-sectional survey and conducted in Henan, China, a populous province with 108.5 million [20], from September 2017 to May 2018. The study population was permanent residents aged 15 to 69 years in 18 provincial cities and 10 provinces directly governing counties under the jurisdiction of Henan Province, which involved those who have accumulated over 6 months of local residence in the past 12 months, regardless of whether they have local household registration. However, the residents who collectively live in military bases, hospitals, prisons, nursing homes, dormitories, and other places were excluded.

A multi-stage cluster sampling method was adopted to select participants. Cluster sampling has a merit that it can fully ensure the consistency of sample structure and population while enhancing the representativeness of samples. The procedure is illustrated in Figure 1. The sample size of each chosen urban resident committee or village committee is calculated by
N = (μ_α_^2^ × p(1 − p))/δ^2^ × *deff*(1)
where α denotes the significance level; $\mu_{\alpha }\;$ is the value of μ when α is equal to 0.05; *p* is the percentage of people with basic health literacy; δ is the maximum permissible error; *deff* represents the design effect of complex sampling adopted to adjust the effectiveness loss due to complex sampling instead of simple random sampling. The monitoring sample rate of residents’ health literacy level in Henan province in 2014 was 8.02% [21]; on that basis, a 95% confidence limit was set, the relative error rate was not greater than 20%, *deff* = 1.5. Based on the results of urban and rural stratification, the invalid questionnaire and the refusal rate was not more than 10%, the minimum monitoring sample size of the provincial municipality was calculated as 3672 people in 17 provincial cities. Likewise, the formula was adopted to calculate the minimum monitoring sample size of 1632 people in each province directly governing county and Jiyuan city. Lastly, the total sample size of Henan province was expected as 3680 × 17 + 1640 × (10 + 1) = 80,600. Those questionnaires with missing values for critical information (address, gender, and age) or health literacy outcome variables were excluded. After data cleaning [13], 78,646 valid questionnaires were analyzed.

### 2.2. Measures and Data Collection

The questionnaire was split into two parts: the first part designed to collect socio-demographic characteristics (e.g., gender, age, education level, and occupation), and the other part assessing the health literacy content based on the “the Chinese Resident Health Literacy Scale” scale developed by the Chinese Ministry of Health. The overall Cronbach’s alpha of the scale was 0.95, and the Spearman–Brown coefficient was 0.94. Cronbach’s alpha of the three dimensions included 0.90 (knowledge), 0.83 (behavior and lifestyle), and 0.85 (skills); thus, the scale exhibits strong psychometric properties with minor measurement invariance [22]. The content consists of 3 aspects (i.e., basic knowledge and concepts, healthy behavior and lifestyle, and health-related skills) and 6 types of health issues (e.g., scientific views of health, prevention of infectious diseases, prevention of chronic diseases, safety and first aid, medical care, as well as health information), which were investigated in households by unified trained investigators. The flow chart of household investigation is given in Figure 2. Health literacy level is the proportion of the population with basic health literacy. The criterion for health literacy is a questionnaire score of 80% or more. This questionnaire consists of 56 questions, among which 50 are included in the calculation of health literacy level, with a total score of 66. If the comprehensive health literacy score reaches 80% or more of the total score (≥53), the respondent will be considered to have basic health literacy. The total score is the sum of all questions of a certain aspect or a certain type of health problem literacy. If the actual score reaches 80% or more of the total score, the person will be considered to exhibit the health literacy of that aspect or that type of health issue. Furthermore, in this study, the health literacy level is the percentage of people having basic health literacy in the total population.

The study complied with the ethical standards of the Ethics Committee of Henan Center for Disease Control and Prevention. All participants completed informed consent forms before being interviewed, in which the purpose of the study was detailed, and a confidentiality agreement of personal information was included.

### 2.3. Statistical Analysis

Basic socio-demographic variables were described by descriptive statistics. Chi-square bivariate tests were performed to determine the group differences (having basic health literacy or not) for all Pthe demographic variables. This cross-sectional survey adopted the method of complex sampling, so the health literacy-related data were weighted according to the sixth national census [13]. Comparisons of the level in different aspects and health issues of health literacy between urban and rural populations were drawn by chi-square tests. Moreover, stepwise logistic regression was conducted to verify whether socio-demographic and health variables are associated with health literacy levels in urban and rural populations. The statistical significance test level was set to 0.05 (two-tailed). All statistical analysis was conducted with SPSS version 21.0 (IBM, Armonk, NY, USA).

## 3. Results

### 3.1. Population Characteristics and Health Literacy Status

The mean age of the respondents was (48.21 ± 13.05) years. In the region, 22.65% were urban populations, and 77.35% were rural populations. The male ratio was 46.57%. Most respondents finished their education in junior middle school. In addition, the respondents were primarily farmers (72.72%). After data weighted adjustment, the level of health literacy in the residents was 10.21%. The level of health literacy in urban populations (16.92%) was higher than that in rural populations (8.09%), and the difference was statistically significant (*p* < 0.001). The level of health literacy was higher in those better educated. All sample characteristics and health literacy status are listed in Table 1.

### 3.2. The Urban–Rural Disparity in Different Aspects and Health Issues of Health Literacy

Figure 3 presents the level of health literacy stratified by 3 different aspects of urban and rural populations. Whether in rural or urban areas, the level of health literacy in healthy behavior and lifestyle was relatively low, and the level of health literacy in basic knowledge and concepts was higher than the level of health literacy in health-related skills. As revealed from the results of chi-square tests, the awareness rate of basic knowledge and concept (34.50% vs. 20.42%), the possession rate of healthy behavior and lifestyle (14.03% vs. 7.33%) and the mastery rate of health-related skills (23.01% vs. 12.66% of urban residents were significantly higher than those in rural residents (*p* < 0.001).

Figure 4 shows the level of health literacy stratified by six types of health issues in urban and rural populations. It is suggested that the health literacy level of prevention of chronic diseases was the lowest, and the health literacy level of safety and first aid was higher than the level of health literacy in other types of health issues in both rural and urban populations. Moreover, a significant difference was identified in the level of health literacy of different types of health issues: scientific views of health (36.84% vs. 23.71%), prevention of infectious diseases (20.59% vs. 13.5%), prevention of chronic diseases (13.68 % vs. 8.07%), safety and first aid (58.87% vs. 44.9%), medical care (21.3% vs. 14.26%), and health information (25.27% vs. 12.68%) between urban and rural residents (*p* < 0.001).

### 3.3. The Urban–Rural Disparity of Risk Factors Associated with Health Literacy

A slight difference was identified in the proportion of men between urban and rural respondents (45.21% vs. 46.24%). An education disparity was identified between urban and rural populations as the urban population had a higher proportion of residents finished high school and above (22.96 vs. 47.82%). Moreover, it also reported that the rural population had a higher proportion of residents engaged in agriculture (73.81 vs. 12.26%).

Since multiple risk factors may be involved, several risk factors should be controlled to simultaneously analyze characteristics associated with health literacy. A multiple logistic regression model was considered in both rural and urban populations. On the whole, the first category of education, occupation, and age groups is applied as a reference. The results of stepwise logistic regression analyses are listed in Table 2.

In urban areas, after all other risk factors were regulated in the logistic regression model, residents with an education level of junior middle school (OR = 1.537, 95% CI (1.154, 2.048)), high school or vocational school (OR = 2.742, 95% CI (2.051, 3.664)), diploma or undergraduate (OR = 5.142, 95% CI (3.816, 6.928)) and postgraduate and above (OR = 10.552, 95% CI (6.738, 16.523)) achieved higher odds to exhibit basic health literacy than those uneducated. Residents who were a teacher (OR = 1.449, 95% CI (1.069, 1.964)) and a medical staff (OR = 2.14, 95% CI (1.553, 2.949)), compared with the civil servant were more likely to exhibit basic health literacy, while residents who were farmers (OR = 0.676, 95% CI (0.508, 0.899)) had 67.6% lower odds to exhibit basic health literacy. Residents aged from 25 to 34 years (OR = 1.422, 95% CI (1.122, 1.803)), 35 to 44 years (OR = 1.688, 95% CI (1.335, 2.134)) and 45 to 54 years (OR = 1.332, 95% CI (1.053, 1.686)) had higher odds to exhibit basic health literacy than those aged 15 to 24 years.

In rural areas, as long as residents educated, they all had higher odds of exhibiting basic health literacy than those uneducated. Moreover, residents who were medical staff (OR = 2.261, 95% CI (1.734, 2.949)), compared with the civil servants, were more likely to exhibit basic health literacy. However, residents who were farmers (OR = 0.672, 95% CI (0.529, 0.853)) were 67.2% less likely to exhibit basic health literacy. Moreover, for age groups, residents aged from 25 to 34 years (OR = 1.224, 95% CI (1.055, 1.421)) and 35 to 44 years (OR = 1.17, 95% CI (1.010, 1.356)) had higher odds of exhibiting basic health literacy than those aged 15 to 24 years, while residents aged 45 to 54 years (OR = 0.846, 95% CI (0.730, 0.981)), 55 to 64 years (OR = 0.716, 95% CI (0.614, 0.836)) and above 65 years (OR = 0.679, 95% CI (0.567, 0.812)) were 84.6%, 71.6% and 67.9%, respectively, less likely to exhibit basic health literacy.

## 4. Discussion

This study was a large-scale survey of public health literacy. The disparity of the level of health literacy and its risk factors among the residents in central China were determined. Moreover, according to the results of this study, the health literacy status was obviously different in urban and rural, indicating a higher level of comprehensive health literacy and different aspects and health issues of health literacy in urban areas with a statistical significance. Though education levels, occupation, and different age groups were associated with health literacy, whereas they were not exactly the same between urban and rural areas.

Results reported that the health literacy level is slightly lower in central China (10.72%) than the national average of 14.18% in 2017 [23] and other investigations in Southern China [11,24]. Though health literacy results are always difficult to compare, low health literacy is still prevalent in China. It was more likely for people who live in the advanced economy regions to exhibit basic health literacy. Besides, the health literacy of residents in central China reported in this study was also less than some developed-countries, (e.g., Germany (47.3–66.4%) [9,25], Korea (39.0%) [26], and the United States [27]. The mentioned data showed that improving the health literacy level of residents in central China is still the focus and difficulty of health education and health promotion.

Results from our survey showed that the level of health literacy in rural populations was obviously lower than that in urban areas in central China, which complies with a study in Iran [28]. The reason may be that rural populations are affected by various factors, which cover underdeveloped economic level, poor basic life implementation, poor medical and health service level, limited access to health information from sources (e.g., primary care providers, specialist doctors, blogs, and magazines) [29], and low awareness of the development of good healthy lifestyle and behavior, thereby causing the generally low level of health literacy among rural residents.

Besides the comprehensive health literacy, three aspects and six health issues of health literacy were also separately assessed and compared in urban and rural populations. Among the three aspects of health literacy, whether in rural or urban areas, residents’ ownership of healthy behavior and lifestyle was the lowest, which complies with the national research results [30] but differs from the results in Shanghai [11]. According to the Knowledge, Attitude/Belief, Practice (KAP) Model [31], the knowledge-to-behavior change is a long process, and the key is to rely on the change of belief and attitude. Accordingly, it is necessary to strengthen residents’ health education and health promotion work, especially in rural areas, so it can help people acquire more health-literacy-related knowledge, strengthen the belief and change attitudes of health-related behavior, as an attempt to promote the formation of healthy lifestyle and behavior. In areas with a relatively developed economy and culture (e.g., Shanghai), residents are more likely to adopt a healthy lifestyle. This also suggests that we should focus on rural areas in the future health literacy promotion, strengthen rural health investment, optimize the allocation of health resources at the grass-roots level, narrow the gap between urban and rural health services, and elevate the level of rural health literacy. While for six health issues of health literacy, differing from the existing study [21], this study found that there is a great disparity with statistical significance between urban and rural populations, probably due to the improvement of China’s health literacy monitoring project and the increasing number of respondents, which makes the results more representative and persuasive than before. Note that the health literacy level of the prevention of chronic diseases is only 8.07% in rural areas. With the change of lifestyle and disease spectrum, the high morbidity and mortality of chronic diseases and the disease burden caused by them continue to threaten human health [32], and the prevalence of common chronic diseases (such as stroke, coronary heart disease and diabetes mellitus) remains high in rural China [33], so a suggestion is that appropriate interventions to improve public health literacy should pay more attention to the prevention of chronic diseases, especially residents living in rural areas. Moreover, just as important, during the epidemic of COVID-19, the health literacy of residents in the prevention of infectious diseases deserves more attention [34,35]. However, the results of this study showed that the health literacy of rural residents in the prevention of infectious diseases is only slightly higher than that of chronic diseases. Therefore, it is recommended that the government and medical practitioners should take this epidemic prevention and control as an opportunity, take the rural population as the focus object, use the advantage of mobile Internet to deepen the health education work of residents’ infectious disease prevention and control, popularize the relevant knowledge and skills of infectious disease protection, and boost the continuous improvement of the long-term mechanism of residents’ scientific quality construction.

As impacted by the significant differences in culture, social background, and economy, the level of health literacy and its risk factors in urban and rural areas may be different. Consistent with previous studies [8,24,36], this study found that health literacy was strongly associated with both demographic characteristics (i.e., age and residence) and socio-economic status (i.e., occupation and education level). Banihashemi et al. attributed the inadequate level of health literacy among rural residents to their low educational status [37]. Likewise, a significant difference in health literacy was found by educational status among rural residents, whereas having a higher level of education might not necessarily lead to better health literacy [29]. From practical perspectives, it is easier to identify residents by occupation than by education level or income [11]. Respondents residing in rural areas who were farmers have lower levels of health literacy than other occupations, and the mentioned characteristics could help identify key populations of health literacy interventions so as to make them access high-quality health care. In agreement with Berens et al., age groups are an important factor associated with health literacy [9], but the results of this study are not exactly the same as theirs. This study found that residents aged from 25 to 34 years, 35 to 44 years and 45 to 54 years all had higher odds to exhibit basic health literacy than those aged 15 to 24 years in urban areas, while residents aged 45 to 54 years, 55 to 64 years were 84.6%, 71.6%, respectively, less likely to exhibit basic health literacy. The reason may be that compared with the rural middle-aged and elderly people, urban middle-aged and elderly people will pay more attention to their own health due to their education level, occupation, and living environment. Thus, it is suggested that health literacy as a determinant of health and social welfare should be focused on with more detail by health decision-makers (e.g., age groups in urban and rural areas).

Some limitations of this study should be considered. First, the data in this study came from a cross-sectional survey, and this confined the interpretation of the results of this study, making it hard to draw causal conclusions. Second, the comparability of the results of this study with other countries might not be well ensured due to the homegrown measures of health literacy and urban–rural differences. Third, this study did not measure the health literacy level of residents aged 70 and over, so the results could not well explain the differences in health literacy between urban and rural elderly people. Future research on health literacy should conduct in old age and explore the role in health disparities. Moreover, health literacy was associated with multiple risk factors and many of the risk factors, but not all of them were included in this study (e.g., economic status and types of chronic diseases that were not analyzed in this study).

## 5. Conclusions

This study suggests that the level of health literacy is relatively low in central China. Moreover, a significant difference was identified of comprehensive health literacy status and different aspects and health issues of health literacy between urban and rural populations, and their risk factors are not the same. The mentioned findings will present the baseline information to develop more effective approaches to enhance the health literacy of rural and urban residents. In this regard, this study suggests that the disparity in fields should be considered in the implementation of health-literacy-related interventions. Moreover, considering the lower health literacy among rural residents compared with their urban counterparts, a reorientation of the health policy-making for Chinese rural areas is also recommended.

## Figures and Tables

**Figure 1 ijerph-17-03848-f001:**
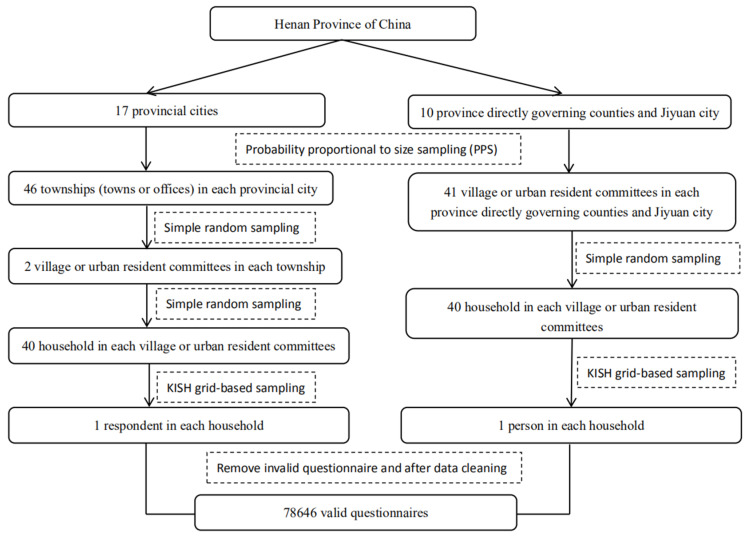
Flow chart for participants sampling in the study on health literacy in a 15 to 69-year-old population, Henan, China, 2016.

**Figure 2 ijerph-17-03848-f002:**
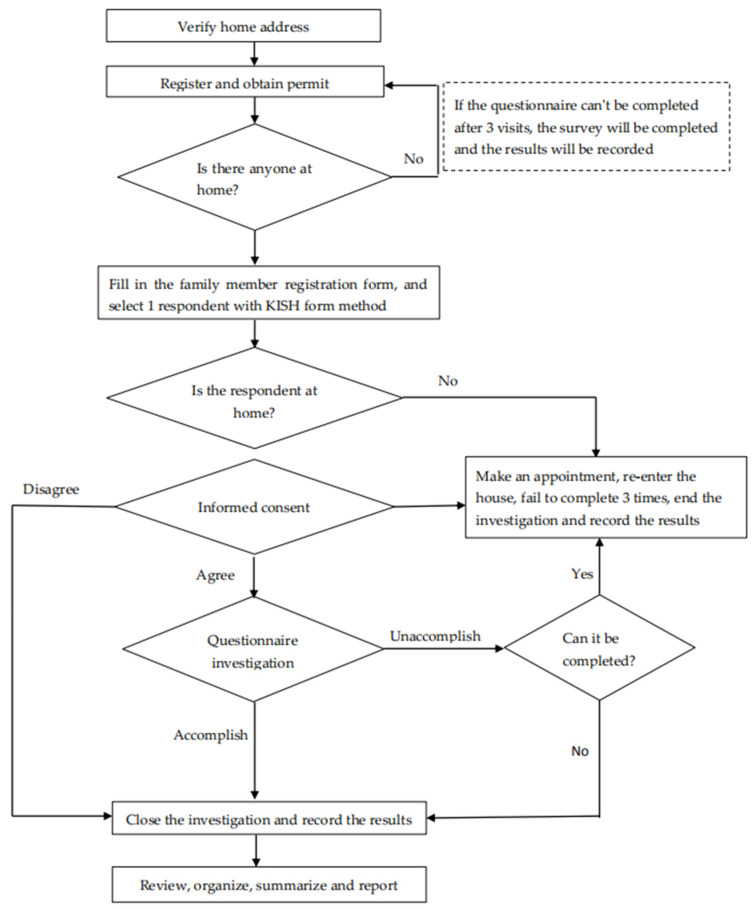
Flow chart for household investigation in the study on health literacy in a 15 to 69-year-old population, Henan, China, 2016. Kish grid-based sampling within each household.

**Figure 3 ijerph-17-03848-f003:**
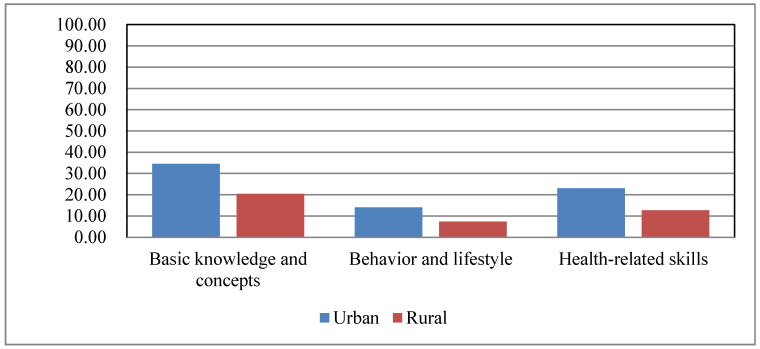
The level of health literacy in 3 different aspects (i.e., basic knowledge and concepts, healthy behavior and lifestyle, and health-related skills) in urban and rural populations.

**Figure 4 ijerph-17-03848-f004:**
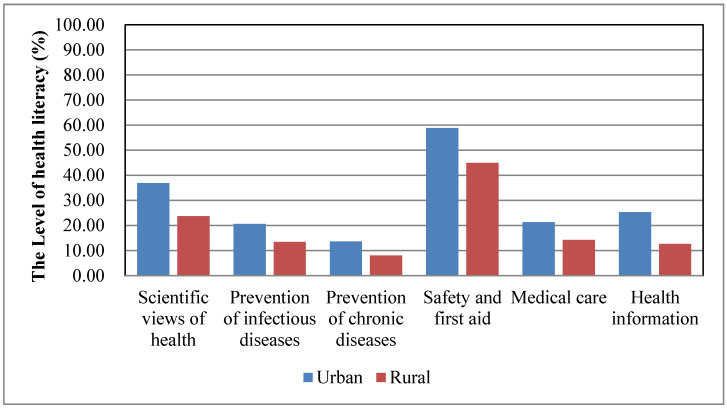
The level of health literacy in 6 types of health problems: scientific views of health, prevention of infectious diseases, prevention of chronic diseases, safety and first aid, medical care, and health information.

**Table 1 ijerph-17-03848-t001:** Sample characteristics and level of health literacy in the participants (*n* = 78,646).

Variable	Category	*n*	%	Level of Health Literacy		χ^2^	*p*
				Sample Rate	Weighted Rate		
Region	Urban	17,813	22.65	15.65	16.92		
	Rural	60,833	77.35	5.77	8.09		
						1826.324	<0.001
Sex	Male	36,625	46.57	8.32	10.10		
	Female	42,021	53.43	7.94	10.31		
						3.800	>0.05
Age groups	15–24	3594	4.57	11.30	11.39		
	25–34	10,303	13.10	14.11	14.88		
	35–44	13,527	17.20	11.67	12.01		
	45–54	24,396	31.02	6.51	6.81		
	55–64	18,254	23.21	5.17	5.34		
	65–69	8572	10.90	3.82	4.10		
						1298.920	<0.001
Education	Uneducated	12,088	15.37	1.91	2.13		
	Primary school	18,254	23.21	3.60	4.16		
	Junior middle school	31,104	39.55	6.38	7.08		
	High school or vocational school	11,718	14.90	14.41	14.65		
	Diploma or undergraduate	5262	6.69	31.35	29.44		
	Postgraduate and above	220	0.28	36.41	38.40		
						5996.713	<0.001
Occupation	Civil servant	645	0.82	22.14	20.71		
	Teacher	1510	1.92	24.20	23.46		
	Medical staff	1258	1.6	35.04	37.78		
	Personnel of other government-sponsored institution	2084	2.65	19.65	18.68		
	Student	1486	1.89	12.81	13.63		
	Farmer	57,191	72.72	4.65	5.83		
	Worker	6064	7.71	13.42	14.64		
	Personnel of other enterprises	3201	4.07	20.42	20.86		
	Others	5207	6.62	15.24	15.68		
						4434.476	<0.001
Total		78,646		8.01	10.21		

**Table 2 ijerph-17-03848-t002:** Odds ratios in favor of having basic health literacy and 95% CI in the stepwise logistic regression.

Risk Factor		Urban		Rural	
		OR (95% CI)	*p*	OR (95% CI)	*p*
Education	Uneducated	Reference			
	Primary school	1.071 (0.780, 1.471)	0.670	1.897 (1.625,2.215)	<0.001
	Junior middle school	1.537 (1.154, 2.048)	0.003	3.164 (2.738, 3.656)	<0.001
	High school or vocational school	2.742 (2.051, 3.664)	<0.001	5.979 (5.136, 6.959)	<0.001
	Diploma or undergraduate	5.142 (3.816, 6.928)	<0.001	10.844 (9.185, 12.803)	<0.001
	Postgraduate and above	10.552 (6.738, 16.523)	<0.001	13.541 (9.749, 18.809)	<0.001
Occupation	Civil servant	Reference			
	Teacher	1.449 (1.069, 1.964)	0.017	1.233 (0.952, 1.595)	0.112
	Medical staff	2.14 (1.553, 2.949)	<0.001	2.261 (1.734, 2.949)	<0.001
	Personnel of other institutions	1.209 (0.910, 1.606)	0.190	1.112 (0.863, 1.433)	0.412
	Student	1.116 (0.742, 1.679)	0.597	0.949 (0.699, 1.289)	0.738
	Farmer	0.676 (0.508, 0.899)	0.007	0.672 (0.529, 0.853)	0.001
	Worker	1.122 (0.851, 1.478)	0.415	1.115 (0.873, 1.424)	0.383
	Other enterprise personnel	1.218 (0.927, 1.600)	0.157	1.246 (0.976, 1.591)	0.077
	Others	1.076 (0.819, 1.415)	0.598	1.176 (0.922, 1.500)	0.191
Age groups	15–24	Reference			
	25–34	1.422 (1.122, 1.803)	0.004	1.224 (1.055, 1.421)	0.008
	35–44	1.688 (1.335, 2.134)	<0.001	1.17 (1.010, 1.356)	0.037
	45–54	1.332 (1.053, 1.686)	0.017	0.846 (0.730, 0.981)	0.027
	55–64	1.156 (0.903, 1.479)	0.251	0.716 (0.614, 0.836)	<0.001
	65–69	1.293 (0.980, 1.707)	0.070	0.679 (0.567, 0.812)	<0.001

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
