# Peer review of "The Urban-Rural Disparity in the Status and Risk Factors of Health Literacy: A Cross-Sectional Survey in Central China"

_ijerph, 2020, doi:10.3390/ijerph17113848_

Round 1
Reviewer 1 Report
General
- Please dont abbreviate Health Literacy -this is not an acceptable abbreviation
- p-values are reported as .05 not 0.05
Abstract
- Health literacy need to be defined
- When results are presented eg. education level - you need to indicate what about education level was significant (eg. higher or lower levels of education
Introduction
- Adequate HL should be defined (P2. L69)
- How does the Chinese HL scale relate to other HL scales and the different types of Health literacy eg. functional - basic? interactive and Critical
Material and Methods
- Were issues of consent of participants under 18 addressed
- More information is needed on the Chinese Resident Health Literacy Scale
- What is the validity and reliability of the HL scale
- The differentiation between the score and a score of 80% or more needs to be consistently identified throughout the paper - this is not clear in the tables. Is 80% and more "basic HL"?
Results
- P5. L144 - overall level of HL - what does this refer to? Is this basic or the 80% or is this the same?
- Table 1
- N should be n
- Please add the Tests conducted to the Table
- Is the HL level the score or the %
- Figures should rather be done as tables with Test and significance - again -Figures appear to be the % - can we have the scores and the %
Author Response
Response to Reviewer 1 Comments
Dear reviewer:
On behalf of my co-authors, we thank you very much for your positive and constructive comments and suggestions on our manuscript entitled “The Urban-rural Disparity in the Status and Risk Factors of Health Literacy: a Cross-Sectional Survey in Central China” (ID: ijerph-787496). We have studied your comments carefully and tried our best to revise our manuscript according to the comments. The followings are your points and our responses:
Point 1: General: Please dont abbreviate Health Literacy -this is not an acceptable abbreviation.
Response 1: Thank you for pointing this out. We have replaced “HL” with “health literacy” in our manuscript.
Point 2: General: p-values are reported as .05 not 0.05
Response 2: Thank you for pointing this out. We have replaced 0.05 with .05 and we also replaced 0.001 with .001.
Point 3: Abstract: Health literacy need to be defined
Response 3: Thank you for your comments. We have added the definition of health literacy: “Health literacy refers to the ability of individuals to access, process, and understand health information to make decisions about treatment and their health in general” in our abstract.
Point 4: Abstract: When results are presented eg. education level - you need to indicate what about education level was significant (eg. higher or lower levels of education)
Response 4: Thank you for your comments. We have corrected the sentence into “The health literacy level was lower in those who had lower levels of education, and there was a significant difference in the level of health literacy among people of different ages and occupations in both urban and rural areas.”
Point 5: Introduction: Adequate HL should be defined (P2. L69)
Response 5: Thank you for your comments. An individual with an adequate level of health literacy refer to he or she has the ability to take responsibility for one’s own health as well as one’s family health and community health. Reference: Sorensen, K.; Van den Broucke S.; Fullam, J.; Doyle, G.; Pelikan, J.; Slonska, Z.; Brand, H. Health literacy and public health: a systematic review and integration of definitions and models. BMC Public Health 2012, 12, 80.
Point 6: Introduction: How does the Chinese HL scale relate to other HL scales and the different types of Health literacy eg. functional - basic? interactive and Critical
Response 6: Thank you for your comments. Functional health literacy (eg.Test of Functional Health Literacy in Adults, TOFHLA) and higher-level health literacy include interactive health literacy and critical health literacy (eg. Critical Health Competence Test, CHC-Test), which are different dimensions of health literacy model based on public health perspective. In 2005, the definition of health literacy first introduced in China adopted the definition of health literacy of the National Institutes of Health, which has been used up to now, and the Chinese Resident Health Literacy Scale is a universal health literacy assessment tool based on the localization of China from the perspective of public health.
Point 7: Material and Methods: Were issues of consent of participants under 18 addressed
Response 7: Thank you for pointing this out. In our study, informed consent was signed by the legal guardian (Usually their parents) of participants younger than 18 years old.
Point 8: Material and Methods: More information is needed on the Chinese Resident Health Literacy Scale
Response 8: Thank you for your comments. We have added more information about the Chinese Resident Health Literacy Scale: “The overall Cronbach’s alpha of the scale was 0.95 and Spearman-Brown coefficient 0.94. Cronbach’s alpha of the three dimensions was as follows: 0.90 (knowledge); 0.83 (behavior and lifestyle); and 0.85 (skills).” in Material and Methods
Point 9: Material and Methods: What is the validity and reliability of the HL scale
Response 9: Thank you for your comments. The overall Cronbach’s alpha of the scale was 0.95 and Spearman-Brown coefficient 0.94. Cronbach’s alpha of the three dimensions was as follows: 0.90 (knowledge); 0.83 (behavior and lifestyle); and 0.85 (skills). With the CFA results, the three-factor model showed slightly better fit than the one-factor model. Correlations among the three factors (knowledge and attitudes; behavior and lifestyle; skills) were 0.96–0.98, so the scale has strong psychometric properties with minor measurement invariance.
Point 10: Material and Methods: The differentiation between the score and a score of 80% or more needs to be consistently identified throughout the paper - this is not clear in the tables. Is 80% and more "basic HL"?
Response 10: Thank you for pointing this out. In our study, we have said that “If the comprehensive health literacy score reaches 80% or more of the total score (≥53), the respondent is judged to having basic health literacy” in Material and Methods. And we have added “and in our study the level of health literacy means that the percentage of respondents who have basic health literacy” in Material and Methods to make it more clear in the tables.
Point 11: Results: P5. L144 - overall level of HL - what does this refer to? Is this basic or the 80% or is this the same?
Response 11: Thank you for pointing this out. They are the same actually. We have replaced “ the overall level of health literacy” with “the level of health literacy”, and we have clarified the relationship between this basic and the 80%.
Point 12: Table 1 : â‘ N should be n; â‘¡ Please add the Tests conducted to the Table; â‘¢ Is the HL level the score or the %
Response 12: Thank you for pointing this out. We have replaced “ N” with “n” in our manuscript; We have added the Tests conducted to the Table 1; Yes, the health literacy level is the %.
Point 13: Figures should rather be done as tables with Test and significance - again -Figures appear to be the % - can we have the scores and the %
Response 13: Thank you for your comments. We agree that Figures should rather be done as tables with Test and significance, but in this study we used Figure just want to visually show the difference of the % between urban and rural health literacy, so we decided to retain our figures, and we think the % is more meaningful than the scores to explain our research results, so we chose to report the %. And we will refer to your opinion in future research to realize the combination of Figures and Tables, thank you so much.
That is all. Thank you very much. We sincerely hope this manuscript will be finally acceptable to be published on IJERPH.

Reviewer 2 Report
Dear autors,
thank you for your interest article, which seems to be an important topic in HL-research. There are some points to note:
In common:
- The english language should be reviewed. There are some sentences which are hard to understand. For example: line 79 can be splitted in two sentences.
- The text is announced with the differences between urban and rural. The discussion starts to explain the focus is about the HL of the population in Central China. It is possible to present both, the focus must be sharpened.
Line 24: Is it about people with(out) education? Question arise, why it is only primary school. For the abstract it would suffice in general to avoid questions
Line 35: Is this a direct quode (HL definition)? Please identify as a dircect quote.
Line 46: Why it is important, how many people lives in China? And: China is compared with other states according to HL research and it it mentioned "public attention on HL rather late". If you set European research (HLS-EU-survey) or other countries China is not as late.
Line 55: Can you mention the used questionnaire?
Method:
Line 85: Why do you conduct a multistage cluster sampling? What are the advantages? You explain the negative aspects (not represantative for population) in the limitation chapter. I am missing the reason for it.
Results:
Line 141: only 1 out of 5 wer from urban populations. Can you describe, if you see an impact of the data and how you handle it. Also, it is described in detail the sampling method. Why you decide for this and not to have a sample 50%/50%.
Line 183: From here, only results are presented which can be seen in table 2. It would be more interest to present other results than results, which are already offered.
Discussion:
Line 211 Comparing the HL results is always difficult. This is mentioned in the limitation, but should be mentioned here.
Line 269: This is not clear. Comparison with young people, who are depending to adults and are in schools (age 15 up to) could not be compared with adults. There is specific research on this age group.
Author Response
Response to Reviewer 2 Comments
Dear reviewer:
On behalf of my co-authors, we thank you very much for your positive and constructive comments and suggestions on our manuscript entitled “The Urban-rural Disparity in the Status and Risk Factors of Health Literacy: a Cross-Sectional Survey in Central China” (ID: ijerph-787496). We have studied your comments carefully and tried our best to revise our manuscript according to the comments. The followings are your points and our responses:
Point 1: In common: The english language should be reviewed. There are some sentences which are hard to understand. For example: line 79 can be splitted in two sentences.
Response 1: Thank you for your comments. Our manuscript has checked by a professional English editing service. It is easy to understand now.
Point 2: In common: The text is announced with the differences between urban and rural. The discussion starts to explain the focus is about the HL of the population in Central China. It is possible to present both, the focus must be sharpened.
Response 2: Thank you for your comments. We agree with you that it is possible to present both, the focus must be sharpened. We have made a change about Discussion in our manuscript to sharpen the differences between urban and rural.
Point 3: Line 24: Is it about people with(out) education? Question arise, why it is only primary school. For the abstract it would suffice in general to avoid questions
Response 3: Thank you for pointing this out. Line 24: It is about people with(out) education, and we agree that for the abstract it would suffice in general to avoid questions. So we replaced “in rural areas, residents who were finished primary school (OR=1.897,95% CI[1.625,2.2150) had higher odds of having basic health literacyHL than residents who were uneducated” with “in rural areas, as long as residents who were educated, they all had higher odds of having basic HL than residents who were uneducated.”
Point 4: Line 35: Is this a direct quode (HL definition)? Please identify as a dircect quote.
Response 4: Thank you for pointing this out. We have identified a direct quode about HL definition in our manuscript.
Point 5: In common: The english language should be reviewed. There are some sentences which are hard to understand. For example: line 79 can be splitted in two sentences.
Response 5: Thank you for your comments. Our manuscript has checked by a professional English editing service. We have corrected errors in grammar and made our manuscript easy to understand.
Point 6: Line 46: Why it is important, how many people lives in China? And: China is compared with other states according to HL research and it it mentioned "public attention on HL rather late". If you set European research (HLS-EU-survey) or other countries China is not as late.
Response 6: Thank you for your comments. China is the most populous country in the world with a population of approximately 1.4 billion. We agree that if we set European research (HLS-EU-survey) or other countries China is not as late, while we just wanted to explain that China's health literacy research started relatively late and developed slowly.
Point 7: Line 55: Can you mention the used questionnaire?
Response 7: Thank you for your comments. The used questionnaire design in 2012 was based on the "Chinese citizens' health literacy basic knowledge and skills " issued by the Chinese Ministry of Health and the "Chinese citizens' health literacy basic knowledge and skills interpretation" compiled by experts organized by the Chinese Ministry of Health, including basic knowledge and concepts, healthy behavior and lifestyle, and health-related skills. The Cronbach coefficient of the questionnaire is 0.931, the Cronbach coefficient of basic knowledge and concepts, healthy behavior and lifestyle, and health-related skills are 0.871, 0.774 and 0.802 respectively. The Chinese Resident Health Literacy Scale in our study is based on the questionnaire in 2012.
Point 8: Method: Line 85: Why do you conduct a multistage cluster sampling? What are the advantages? You explain the negative aspects (not represantative for population) in the limitation chapter. I am missing the reason for it.
Response 8: Thank you for your comments. We conducted a multistage cluster sampling in Henan Province. The advantage of multi-stage stratified sampling is that it can fully ensure the consistency of sample structure and population, and improve the representativeness of samples. While in the limitation chapter we explained our sample in Henan Province could not well represent the health literacy level of Central China. Maybe it was our unclear narration that made you a little confused. We have rewritten it in the limitation chapter.
Point 9: Results: Line 141: only 1 out of 5 wer from urban populations. Can you describe, if you see an impact of the data and how you handle it. Also, it is described in detail the sampling method. Why you decide for this and not to have a sample 50%/50%.
Response 9: Thank you for pointing this out. The reason why only 1 out of 5 wer from urban populations is related to the proportion of urban and rural population in Henan Province.The results of the sixth census of China show that the total population of Henan Province at the end of 2010 is 99.67 million, including 37.58 million urban population, accounting for 37.7% of the total population, and 62.09 million rural population, accounting for 62.3% of the total population. We conducted a multistage cluster sampling in Henan Province because the advantage of multi-stage stratified sampling is that it can fully ensure the consistency of sample structure and population, and improve the representativeness of samples.
Point 10: Results: Line 183: From here, only results are presented which can be seen in table 2. It would be more interest to present other results than results, which are already offered.
Response 10: Thank you for your comments. We agree that it would be more interest to present other results. It is a pity we did not carry out the analysis because the data of other demographic factors (such as economic level) which were missing too much. We would pay more attention to it in our future research, thank you very much.
Point 11: Discussion: Line 211 Comparing the HL results is always difficult. This is mentioned in the limitation, but should be mentioned here.
Response 11: Thank you for pointing this out. We decided to accept your proposal, and we have added “Though comparing the HL results is always difficult, this suggested that low HL is still prevalent in China, it was more likely for people to have basic HL who live in the advanced economy regions” in Line 211.
Point 12: Discussion: Line 269: This is not clear. Comparison with young people, who are depending to adults and are in schools (age 15 up to) could not be compared with adults. There is specific research on this age group.
Response 12: Thank you for pointing this out. We agree with you that there should has a specific research on this young age group, while in our discussion we just compared the results with the study called “Health literacy among different age groups in Germany: results of a cross-sectional survey” which also had young people (age 15 up to), maybe it was reasonable.
That is all. Thank you very much. We sincerely hope this manuscript will be finally acceptable to be published on IJERPH.

Reviewer 3 Report
The authors conducted a cross-sectional study, in which 78646 participants were selected from a populous province in Central China utilizing a multi-stage random sampling design. They found the level of Health literacy (HL) in Central China was severely lower than the national average, and a big gap was found between urban and rural population. There is a significant difference in different aspects and health issues of HL between urban and rural populations. Education level, occupation, and different age groups were associated with HL in both urban and rural areas. In rural areas, residents who were finished primary school (OR=1.897,95% CI[1.625,2.2150) had higher odds of having basic HL than residents who were uneducated. Also, among rural area, compared to those who were 15 to 24 years old, residents who were 45 to 54 years old (OR=0.846,95% CI [0.730, 0.981]), 55 to 64 years old (OR=0.716,95% CI[0.614, 0.836]) and above 65 (OR=0.679, 95% CI[0.567, 0.812]) were 84.6%, 71.6%, 67.9%, respectively, less likely to have basic HL. They concluded that in view of the lower HL among rural residents, a reorientation on the health policy-making for Chinese rural areas is recommended. Urban-rural disparity about HL risk factors should be taken into consideration when implementing HL promotion intervention.
Although the idea of urban-rural disparity is not novel, and is well discussed in lots of studies, this research still provides evidence of the gap of HL between urban and rural areas, and between different education levels, occupations, as well as age groups. By using a large sample and a sound sampling method, the results of this research are robust. A minor revision is needed for the manuscript before its publication.
- There are many grammar and spelling errors throughout the manuscript (such as "suggest" in line 30 and "has" in line 37). Please revise the manuscript to improve the readability.
- The style of references is incorrect. Please make changes for the references to conform to the journal’s style.
Author Response
Response to Reviewer 3 Comments
Dear reviewer:
On behalf of my co-authors, we thank you very much for your positive and constructive comments and suggestions on our manuscript entitled “The Urban-rural Disparity in the Status and Risk Factors of Health Literacy: a Cross-Sectional Survey in Central China” (ID: ijerph-787496). We have studied your comments carefully and tried our best to revise our manuscript according to the comments. The followings are your points and our responses:
Point 1: There are many grammar and spelling errors throughout the manuscript (such as "suggest" in line 30 and "has" in line 37). Please revise the manuscript to improve the readability. 

Response 1: Thank you for your comments. Our manuscript has checked by a professional English editing service. It is easy to understand now.
Point 2: The style of references is incorrect. Please make changes for the references to conform to the journal’s style.
Response 2: Thank you for pointing this out. We have made changes for the references to conform to the journal’s style in our manuscript.
That is all. Thank you very much. We sincerely hope this manuscript will be finally acceptable to be published on IJERPH.

Reviewer 4 Report
Language
Proofreading should be done to correct errors in grammar. English must be improved. Some of these are listed below:
Line 15: urban and rural public
Line 19: severely lower
Line 37: public Health. Studies has shown
Line 44: healthcare
Line 45-46: the sentence is not clear
Line 71: is not clear and should be more specific
Line 143: majority of the population was farmer
A brief summary
The aim of the paper is to assess the status of HL and to explore the sociodemographic differences among urban and rural population in Henan province, in Central China. The study shows results of large scale survey in big population.
The aim of this paper has been achieved by the applied analysis and results.
The conclusions based on the results of this study gives suggestions for public health policy in Central China to reduce the gab in HL status between rural and urban pollution.
The strengths of this study is large study sample and the specific description of the sampling design in big populations. The presentation of “the flow chart of household investigation” has educative value.
Broad Comments
The title seems to be not clear and not specific enough. This study is about the HL gab in rural and urban populations in Central China. From the public health perspectives sociodemographic factors are difficult to change by public health policies. I would suggest: “The disparity/gab in the HL status between urban and rural population in Central China”
It’s a pity, and it is an important weakness of this study, that sex (men and females) differences is not consider in the logistic regression analysis, with regard to urban and rural regions.
Specific comments
Abstract
Line 15-19: The aim, objective of the study should be defined before study sample.
The aim should be written as it is written in line 73.
There is lack of short description of the Method, including design and applied tools (questionnaire).
Line 24-25 OR should written be in the end of the part of the sentence
Line 32 Keywords: to make more specific and clear I suggest “urban and rural population”, and rather “sociodemographic factors”
Introduction
Line 49-53 The reference [12] is older than reference [11], so I suggest to add them together
Line 48: Anglo-American Region or Canada as geographical region usually refers specifically to the United States and Canada
Line 50-51: the title of the program should be the same as in the references
Materials and Methods
Line 99: Please explain exclusion criteria, specify what kind of “important personal information”
Line 106: Please specify all sociodemographic variables
There is lack of definition of “personnel of other institutions” and “other enterprise personnel”, examples or characteristics of occupations belonging to these groups. Does these occupations categories applied any occupational status or level, degree.
Results
Line 151-153 is not clear, please rewrite this sentence.
Figures 3 is not clear entitled.
The description of the logistic regression analysis does not explains, why “civil servants” and “age group of 15-24 years” are the references. Why, for example, “medical staff” is not a reference group? I suggest that in Occupation: “Others” should be excluded from the analysis, since this is perhaps “unknown” group, if not, please describe who they are. Another question regards farmers in urban population, please inform how many farmers were in this urban population.
Discussion
Line 229: The reference to KAP model is missed.
Author Response
Response to Reviewer 4 Comments
Dear reviewer:
On behalf of my co-authors, we thank you very much for your positive and constructive comments and suggestions on our manuscript entitled “The Urban-rural Disparity in the Status and Risk Factors of Health Literacy: a Cross-Sectional Survey in Central China” (ID: ijerph-787496). We have studied your comments carefully and tried our best to revise our manuscript according to the comments. The followings are your points and our responses:
Point 1: Language: Proofreading should be done to correct errors in grammar. English must be improved. Some of these are listed below: Line 15: urban and rural public; Line 19: severely lower; Line 37: public Health. Studies has shown; Line 44: healthcare; Line 45-46: the sentence is not clear; Line 71: is not clear and should be more specific; Line 143: majority of the population was farmer
Response 1: Thank you for your comments. Our manuscript has checked by a professional English editing service. We have corrected errors in grammar in our manuscript and make the sentences clear.
Point 2: A brief summary: The aim of the paper is to assess the status of HL and to explore the sociodemographic differences among urban and rural population in Henan province, in Central China. The study shows results of large scale survey in big population.
The aim of this paper has been achieved by the applied analysis and results.
The conclusions based on the results of this study gives suggestions for public health policy in Central China to reduce the gab in HL status between rural and urban pollution.
The strengths of this study is large study sample and the specific description of the sampling design in big populations. The presentation of “the flow chart of household investigation” has educative value.
Response 2: Thank you for your comments and affirmation.
Point 3: Broad Comments: The title seems to be not clear and not specific enough. This study is about the HL gab in rural and urban populations in Central China. From the public health perspectives sociodemographic factors are difficult to change by public health policies. I would suggest: “The disparity/gab in the HL status between urban and rural population in Central China”
Response 3: Thank you for your comments. We agree that from the public health perspectives sociodemographic factors are difficult to change by public health policies. However, exploring the influence of demographic factors on health literacy helps to distinguish the key groups of health education and health promotion, so we decided to use our original title. Thank you very much for your suggestion about the title.
Point 4: Broad Comments: It’s a pity, and it is an important weakness of this study, that sex (men and females) differences is not consider in the logistic regression analysis, with regard to urban and rural regions.
Response 4: Thank you for pointing this out. In our sample, Chi square test in Table1 showed that there is no significant difference in health literacy level between men and females, the same were true of rural and urban populations. Since multiple risk factors may be involved, several risk factors must be controlled for simultaneously analyzing characteristics associated with health literacy. And this is the reason why we did not consider gender in the logistic regression analysis.
Point 5: Abstract: Line 15-19: The aim, objective of the study should be defined before study sample.
Response 5: Thank you for your comments. The aim, objective of the study have been defined before study sample: “we wanted to assess the status of HL and to explore the differences of its possible determinants such as socio-economic factors among urban and rural population in Henan province, China, so in a cross-sectional study, 78646 participants were selected from a populous province in Central China utilizing a multi-stage random sampling design and the Chinese Resident Health Literacy Scale was used to measure the health literacy of the respondents”
Point 6: Abstract: The aim should be written as it is written in line 73.
Response 6: Thank you for your comments. We have written the aim in Abstract according line 73 in our manuscript.
Point 7: Abstract: There is lack of short description of the Method, including design and applied tools (questionnaire).
Response 7: Thank you for your comments. We have added a short description of the Method which said “Against this background, in a cross-sectional study, 78646 participants were selected from a populous province in Central China utilizing a multi-stage random sampling design, and the Chinese Resident Health Literacy Scale was used to measure the health literacy of the respondents.”
Point 8: Abstract: Line 24-25 OR should written be in the end of the part of the sentence
Response 8: Thank you for your comments. We agree with you that OR should written be in the end of the part of the sentence.
Point 9: Abstract: Line 32 Keywords: to make more specific and clear I suggest “urban and rural population”, and rather “sociodemographic factors”
Response 9: Thank you for your comments. I'm sorry we don't quite understand you. "Social digital factors" was not appeared in our keywords, so we have not modified the keywords.
Point 10: Introduction: Line 49-53 The reference [12] is older than reference [11], so I suggest to add them together
Response 10: Thank you for your comments. We have adopted your suggestion to add them together in our manuscript.
Point 11: Introduction: Line 48: Anglo-American Region or Canada as geographical region usually refers specifically to the United States and Canada
Response 11: Thank you for pointing this out. We have replaced “Anglo-American Region” with “the United States”
Point 12: Introduction: Line 50-51: the title of the program should be the same as in the references
Response 12: Thank you for pointing this out. We have changed the title of the program into “the First Chinese Residents Health Literacy Monitoring Program” in our manuscript.
Point 13: Materials and Methods: Line 99: Please exclusion criteria, specify what kind of “important personal information”
Response 13: Thank you for your comments. Those questionnaires with missing values for important information included questionnaire code, address, gender and age were excluded. We have explained it in our manuscript.
Point 14: Materials and Methods: Line 106: Please specify all sociodemographic variables
Response 14: Thank you for your comments. In our study, we did not carry out the analysis because the data of other demographic factors (such as economic level) which were missing too much, so we did not specify this sociodemographic variables in Materials and Methods.
Point 15: Materials and Methods: There is lack of definition of “personnel of other institutions” and “other enterprise personnel”, examples or characteristics of occupations belonging to these groups. Does these occupations categories applied any occupational status or level, degree.
Response 15: Thank you for your comments. The division of occupations in our study is based on the National Program of China's Health Literacy. In China, government-sponsored institution refer to social service organizations organized by state organs or other organizations with state-owned assets and engaged in education, scientific research, culture, health, sports, press and publication, radio and television, social welfare, disaster relief and other activities for the purpose of public welfare. “Personnel of other institutions” refer to “Personnel of other government-sponsored institution”, Civil servants, Teachers and Medical staffs are all staff of personnel of government-sponsored institution. And personnel of other government-sponsored institution refer to personnel of government-sponsored institution other than teachers and doctors. While an enterprise unit is generally a productive unit that is responsible for its own profits and losses. In our study, “other enterprise personnel” refer to personnel of enterprise units other than workers.
Point 15: Results: Line 151-153 is not clear, please rewrite this sentence.
Response 16: Thank you for your comments. We have rewritten this sentence as “Whether in rural or urban areas, the level of health literacy in healthy behavior and lifestyle is relatively low, and the level of health literacy in basic knowledge and concepts is higher than the level of health literacy in health-related skills”
Point 17: Results: Figures 3 is not clear entitled.
Response 17: Thank you for your comments. We have renamed Figures 3 to “The level of health literacy in 3 different aspects (basic knowledge and concepts, healthy behavior and lifestyle, and health-related skills) in urban and rural populations” to make it more appropriate in our manuscript.
Point 18: Results: The description of the logistic regression analysis does not explains, why “civil servants” and “age group of 15-24 years” are the references. Why, for example, “medical staff” is not a reference group? I suggest that in Occupation: “Others” should be excluded from the analysis, since this is perhaps “unknown” group, if not, please describe who they are. Another question regards farmers in urban population, please inform how many farmers were in this urban population.
Response18: Thank you for your comments. We have added the explanation of the references in the manuscript. As our response in Point 15: The division of occupations in our study is based on the National Program of China's Health Literacy. In China, government-sponsored institution refer to social service organizations organized by state organs or other organizations with state-owned assets and engaged in education, scientific research, culture, health, sports, press and publication, radio and television, social welfare, disaster relief and other activities for the purpose of public welfare. “Personnel of other institutions” refer to “Personnel of other government-sponsored institution”, Civil servants, Teachers and Medical staffs are all staff of personnel of government-sponsored institution. And personnel of other government-sponsored institution refer to personnel of government-sponsored institution other than teachers and doctors. While an enterprise unit is generally a productive unit that is responsible for its own profits and losses. In our study, “other enterprise personnel” refer to personnel of enterprise units other than workers. And “Others” refer to people other than students, farmers, Personnel of government-sponsored institutions and enterprises, such as beggarsï¼›In our study, 33% of the urban population are farmers.
Point 19: Discussion: Line 229: The reference to KAP model is missed.
Response 19: Thank you for pointing this out. We have added the reference to KAP model “32. Cleary, A.; Dowling, M. Knowledge and attitudes of mental health professionals in Ireland to the concept of recovery in mental health: a questionnaire survey. J Psychiatr Ment Hlt 2009, 16, 539-545.”
That is all. Thank you very much. We sincerely hope this manuscript will be finally acceptable to be published on IJERPH.

Round 2
Reviewer 4 Report
Language
Proofreading should be done to correct minor errors in English. Some of these are listed below:
Line 16: public. Should be: population or people or residents.
Line 17 and 78: rarely known. Should be: little known.
Line 19: For this end, in a cross-sectional study. Should be: A cross-sectional study.
Line 21-22: Besides. Moreover. Should be deleted.
Line 32: Given. Should be: Regarding or Considering.
Line 238: lack of attention. According to the health education terminology should be: low awareness.
A brief summary
The aim of the paper was to assess the status of HL and to explore the sociodemographic differences among urban and rural population in Henan province in central China. The study shows results of large scale survey in big population. The aim of this paper has been achieved by the correctly applied analysis and results.
The conclusions based on the findings of this study contain worthwhile suggestions for public health policy in central China to reduce the gab in HL status between rural and urban pollution regarding the socioeconomic status differences.
Broad comments
The revised version of the manuscript has been significantly improved and has got much higher quality and important impact to the field.
Discussion part is broad, significant, and interesting to the reader.
During present time of the epidemic of COVID-19, this study brings an important findings and conclusions about the need to implement the HL education and promotion, including the prevention of infectious diseases, among all low socioeconomic status residents, especially in the rural areas in China.
Specific comments
Abstract
Line 25,27: No need to write (p value)
Discussion
Line 24: Regarding Knowledge, Attitude / Belief, Practice (KAP) Model in HL education I would suggest to add more influential public health publications, for example: World Health Organization, Regional Office for the Eastern Mediterranean. (2012). Health education: theoretical concepts, effective strategies and core competencies: a foundation document to guide capacity development of health educators. https://apps.who.int/iris/handle/10665/119953
Author Response
Response to Reviewer 4 Comments
Dear reviewer:
On behalf of my co-authors, we thank you very much for your positive and constructive comments and suggestions on our manuscript entitled “The Urban-rural Disparity in the Status and Risk Factors of Health Literacy: a Cross-Sectional Survey in Central China” (ID: ijerph-787496). We have studied your comments carefully and tried our best to revise our manuscript according to the comments. The followings are your points and our responses:
Point 1: Language:Proofreading should be done to correct minor errors in English. Some of these are listed below:
- Line 16: public. Should be: population or people or residents.
- Line 17 and 78: rarely known. Should be: little known.
- Line 19: For this end, in a cross-sectional study. Should be: A cross-sectional study.
- Line 21-22: Besides. Moreover. Should be deleted.
- Line 32: Given. Should be: Regarding or Considering.
- Line 238: lack of attention. According to the health education terminology should be: low awareness.
Response 1: Thank you for your comments. We have checked our manuscript again to correct minor errors in English, including the following: â‘ We have replaced “public” with “population”. â‘¡We have replaced “rarely known” with “little known”. â‘¢ We have replaced “For this end, in a cross-sectional study” with “A cross-sectional study”. â‘£ We've deleted “Besides” and “Moreover”. ⑤We have replaced “Given” with “Considering”.â‘¥ We have replaced “lack of attention to” with “low awareness of”.
Point 2: A brief summary: The aim of the paper was to assess the status of HL and to explore the sociodemographic differences among urban and rural population in Henan province in central China. The study shows results of large scale survey in big population. The aim of this paper has been achieved by the correctly applied analysis and results.
The conclusions based on the findings of this study contain worthwhile suggestions for public health policy in central China to reduce the gab in HL status between rural and urban pollution regarding the socioeconomic status differences.
Response 2: Thank you for your summary and affirmation.
Point 3: Broad Comments: The revised version of the manuscript has been significantly improved and has got much higher quality and important impact to the field.
Discussion part is broad, significant, and interesting to the reader.
During present time of the epidemic of COVID-19, this study brings an important findings and conclusions about the need to implement the HL education and promotion, including the prevention of infectious diseases, among all low socioeconomic status residents, especially in the rural areas in China.
Response 3: Thank you very much for your affirmation and comments. In the future, we will make interventions to improve the health literacy of vulnerable groups in rural areas according to the results of this study.
Point 4: Specific comments:Abstract Line 25,27: No need to write (p value)
Response 4: Thank you for pointing this out. We have deleted the p value.
Point 5: Discussion:Line 24: Regarding Knowledge, Attitude / Belief, Practice (KAP) Model in HL education I would suggest to add more influential public health publications, for example: World Health Organization, Regional Office for the Eastern Mediterranean. (2012). Health education: theoretical concepts, effective strategies and core competencies: a foundation document to guide capacity development of health educators. https://apps.who.int/iris/handle/10665/119953
Response 5: Thank you for your comments and examples. We have received your suggestion to add “World Health Organization, Regional Office for the Eastern Mediterranean. (2012). Health education: theoretical concepts, effective strategies and core competencies: a foundation document to guide capacity development of health educators. https://apps.who.int/iris/handle/10665/119953” as a reference.
That is all. Thank you very much. We sincerely hope this manuscript will be finally acceptable to be published on IJERPH.
